# A Systematic Review of Current Applications of Fecal Microbiota Transplantation in Horses

**DOI:** 10.3390/vetsci10040290

**Published:** 2023-04-13

**Authors:** Maimaiti Tuniyazi, Wenqing Wang, Naisheng Zhang

**Affiliations:** 1Department of Clinical Veterinary Medicine, College of Veterinary Medicine, Jilin University, Changchun 130062, China; mmttn18@mails.jlu.edu.cn; 2School of Veterinary Science, The University of Queensland, Gatton, QLD 4343, Australia

**Keywords:** equine, fecal microbiota transplantation, gut microbiota

## Abstract

**Simple Summary:**

Fecal microbiota transplantation (FMT) is increasingly recognized in veterinary medicine as a potential treatment for various gastrointestinal disorders in horses. The primary objective of FMT is to restore the healthy gut microbiota in horses by transferring fecal material from a healthy donor to a recipient. The mechanism of FMT remains unclear, but it is hypothesized to work by introducing a diverse and balanced microbial community into the recipient’s gut, which can then compete with and replace harmful pathogens and promote a healthy gut environment. Therefore, in order to better understand current application of FMT in horses, we conducted this systematic review based on the existing literature. According to the research findings, we discussed the key factors that may influence the efficacy and safety, as well as future application, of FMT in horses.

**Abstract:**

Fecal microbiota transplantation (FMT) is a technique involving transferring fecal matter from a healthy donor to a recipient, with the goal of reinstating a healthy microbiome in the recipient’s gut. FMT has been used in horses to manage various gastrointestinal disorders, such as colitis and diarrhea. To evaluate the current literature on the use of FMT in horses, including its efficacy, safety, and potential applications, the authors conducted an extensive search of several databases, including PubMed, MEDLINE, Web of Science, and Google Scholar, published up to 11 January 2023. The authors identified seven studies that met their inclusion criteria, all of which investigated the FMT application as a treatment for gastrointestinal disorders such as colitis and diarrhea. The authors demonstrated that FMT was generally effective in treating these conditions. However, the authors noted that the quality of the studies was generally suboptimal and characterized by small sample sizes and a lack of control groups. The authors concluded that FMT is a promising treatment option for certain gastrointestinal disorders in horses. Nevertheless, more research is required to determine the optimal donor selection, dosing, and administration protocols, as well as the long-term safety and efficacy of FMT in horses.

## 1. Introduction

The horse gastrointestinal tract is a home for heavily populated microorganisms (bacteria, fungi, and viruses) which are collectively known as microbiota. Advances in RNA-sequencing technology (e.g., 16S rRNA) revealed that every part of a horse’s body is colonized with a unique microbial community. The gut microbiota has the most abundant and diverse microbial population compared with other organs and tissue. The primary function of equine gut microbiota is food digestion and nutrient absorption [1]. However, recent research has shown that the gut microbiota also plays an important role in the host’s normal physiological functions. For example, when the gut microbiota is disrupted (dysbiosis) by various factors [2], a number of gastrointestinal diseases occur, such as colitis [3,4] and diarrhea [5]. Therefore, keeping the gut microbiota healthy is vital for the overall well-being of horses.

Fecal microbiota transplantation (FMT) has been extensively studied since its approval for treating human *Clostridium difficile* infections (CDI) by the US Food and Drug Administration in 2013 [6]. Although FMT has gained increasing attention in veterinary medicine as a potential therapy for various gastrointestinal disorders in horses [7,8], our understanding of FMT is still limited. The aim of FMT is to re-establish the healthy gut microbiota in horses by transferring fecal material from a healthy donor to a recipient. The mechanism of FMT action remains elusive, but it is believed to function by introducing a diverse and balanced microbial community into the recipient’s gut, which can then compete with and displace harmful pathogens, leading to a healthier gut environment.

Despite the growing interest and promising results, FMT has not yet become the mainstream treatment option in equine clinics, even though the underlying condition is clearly intestine-based and FMT could offer the most direct and cost-effective approach. The main barrier is safety concerns. Therefore, in this systematic review, we scrutinized the current literature on the use of FMT in horses, including its efficacy and safety.

## 2. Method

### 2.1. Data Sources and Search Strategy

This systematic review was conducted according to the guidelines of the Preferred Reporting Items for Systematic Review and Meta-Analysis Protocol (PRISMA).

A systematic search was performed in four databases (PubMed, MEDLINE, and Web of Science, and for gray articles, Google Scholar limited to the first 10 pages) on 11 January 2023. Databases were searched for fecal microbiota transplantation with the following alternative terms: “fecal”, “faecal”, “microbiota”, “microbiome”, “microflora”, “feces”, “faeces”, “stool”, “stool flora”, “stool microbiota”, “stool bacteria”, “fecal flora”, and “faecal flora”; individually combined with the alternative terms of transplant: “transplant”, “transfusion”, and “transplantation”. These terms were then searched in combination. Then, the results were combined with various equine alternative terms (“horse”, “equine”, “pony”, “mare”, “foal”, “filly”, “gelding”, or “colt”) and combined by the Boolean term “AND”.

After the literature search, all articles were independently reviewed by the two authors (MT and WW).

Literature inclusion criteria: (1) the study population included any horses receiving FMT treatment for any symptoms; (2) the study types were clinical trial, case report, and observational studies; (3) no language limits were used.

Literature exclusion criteria: (1) duplicate publications, conference publications, editorials, and reviews; (2) abstracts, and no full text available.

### 2.2. Data Extraction

From each study, the following information was extracted: author, year of publication, geographic location of the study, sample size, patient characteristics, frequency of fecal transplant therapy, methods for preparing fecal solution, and taxonomy of equine gut microbiota pre-and-post-FMT treatment.

## 3. Results

### 3.1. Study Selection

A total of 561 articles were identified through the database search, and one article was retrieved through cross-reference. After removing duplicates, 237 articles were initially screened by title and abstract. At this step, 218 articles were excluded based on the study design (n = 129), study population (n = 57), and study type (n = 32). A total of 19 articles underwent full-text review. Then, 12 articles were removed as they were conference abstracts or posters, with no availability of full text (n = 12).

### 3.2. Study Characteristics

As shown in Figure 1, a total of seven studies were included in the qualitative synthesis. Characteristics of the seven articles are presented in Table 1. Of these studies, 29% were clinical trials (n = 2), and 71% were observational studies (n = 5). The usage of FMT varied among the studies, with three investigating diarrhea, two investigating antibiotic-induced intestinal dysbiosis, one investigating colitis, and one investigating free fecal water.

A total of 75 horses were enrolled in the included studies, with a mean of 11 per study (sample sizes ranged from 4 to 22). Among them, there were a total of 33 horses in studies relating to diarrhea, 18 for antibiotic-induced dysbiosis, 4 for colitis, and 20 for free fecal water.

Studies were conducted in the US (n = 2) [8,10], Canada (2) [11,12], the Netherlands (n = 1) [13], Japan (n = 1) [14], and Brazil (n = 1) [9].

### 3.3. Methodology of FMT Process

The process for FMT administration was the same among studies: delivered via nasogastric tube. However, the FMT process for obtaining feces and preparing fecal solutions varied between studies.

#### 3.3.1. Fecal Collection

Fecal samples were collected freshly per rectum from each donor horse (n = 4) [9,10,12,13]. The amount of feces ranged from 0.5 kg to 2 kg, with an approximate average of 1.12 kg per horse.

In two studies, the feces were not collected directly from the rectum, but were instead collected after being dropped on the straw bed [14], or collected using a fecal collector [11]. Approximately 10 kg of feces were obtained by an overnight-kept fecal collector [11]. However, the amount of feces is not described precisely in the other study which merely stated ‘the feces were collected within 2 h in several times’ [14].

One study did not specify the method for collecting feces from the donor horse [8], but that it was freshly used for FMT. Out of the seven studies included, only one study reported using centrifugation of the fecal solution [11], while the remaining studies did not go through extra handling of the fecal material.

#### 3.3.2. Fecal Preparation

Of all included studies, fecal solutions were prepared aerobically, and feces were exposed to oxygen during the obtaining and handling procedures.

Out of the seven included studies, five prepared the inoculum using freshly collected feces. Briefly, the collected stool samples were mixed with (n = 1) [8] or without a mixer (n = 4) [9,10,12,13] in water (n = 3) [8,10,12], 10% sodium bicarbonate solution (n = 1) [9], or non-sterile saline (n = 1) [13]. The amount of liquid used to prepare FMT varied between 4 L to 5 L per horse. The liquid was warm or warmed up before administering to the horses.

One study prepared FMT using frozen feces [14]. After collecting, the feces were immediately stored at −20 °C, then at −80 °C. Before use, the frozen feces were thawed with 1 L of warm water, mixed thoroughly, and administered to the horses.

Another study used both fresh and frozen feces [11]. Fresh FMT was conducted by mixing 1.6 kg of stool in 3.2 L of water. Frozen stool suspension was prepared by using 2 L of water and 1 kg of fresh feces. The mixture was centrifuged at 24,470× *g* for 30 min; after the supernatant was discarded, the remainder was resuspended in 400 mL of 10% glycerol in 0.9% saline, then stored in −80 °C until use.

#### 3.3.3. Donor Selection and Screening

As shown in Table 2, all of the seven studies conducted clinical examinations before stool collection. Although these examinations varied among the studies, the main purpose was to ensure the donors had a healthy gut microbiota and were free from antimicrobial therapy for at least three months. Five studies did not specify the exclusion criteria before collecting feces, but were rather based on the recorded information [9,11,12,13,14]. Two studies excluded donor horses based on specific criteria [8,10], such as colic, diarrhea, transport, medical, or probiotic interventions.

After fecal collection, five studies reported further examinations of the stool. Although the examinations were different between studies [8,10,11,12,14], they included coronavirus, *Clostridium difficile* toxins A and B, *Clostridium perfringens* antigens, *Lawsonia intracellularis*, *Neorickettsia risticii*, *Salmonella* sp., and fecal egg count. In contrast, two studies did not detail any examinations after collecting stools [9,13].

### 3.4. The Efficacy and Safety of FMT in Horses

Of all the seven studies, 46 horses were treated with FMT, whereas three of them died at the end. The overall successful/survival rate was 93.48%. Among them, four colitis horses were treated by FMT with a 75% success rate, and 23 diarrhea horses were treated by FMT with a 91.3% success rate; one study found that FMT did not have any significant effect on horses with free fecal water, resulting in a 0% success rate [13]. Two studies examined the efficacy of FMT after the gut microbiota was experimentally disrupted by antibiotics such as metronidazole [14] and trimethoprim sulfadiazine [11]. Therefore, they were not included in the cure rate estimating process.

### 3.5. The Effect of FMT on Gut Microbiota

As shown in Table 3, six out of seven studies conducted fecal microbiota analysis before and after FMT administration. Among them, four studies indicated significant dysbiosis of gut microbiota compositions before FMT. In diarrhetic horses, the fecal microbiota inhibited lower α-diversity and greater beta *β*-diversity [8,10], or increased Lactobacillales order and the genera *Lactobacillus*, *Intestinimonas*, and *Streptococcus* [12]. Di Pietro et al. indicated that after inducing intestinal dysbiosis using trimethoprim sulfadiazine, the gut microbiota showed higher abundance of the genus *Intestinimonas*, *unclassified Lactobacillales*, *Lactobacillus*, and *Streptococcus* compared to controls [11].

After FMT treatments, four studies found that there were no significant differences in the composition of the gut microbiota compared to controls or before FMT treatment [11,12,13,14]. However, the clinical sypmtoms were allevated after FMT. Two studies showed that the horses receiving FMT had similar fecal microbiota compositions to their donors [8,10], as indicated by the lower mean UniFrac distance.

## 4. Discussion

Fecal microbiota transplantation (FMT) has emerged as a promising treatment option for various gastrointestinal disorders in horses. This systematic review provides an overview of the current applications of FMT in horses and highlights the key findings from the available literature. These studies have identified important aspects of FMT in horses, including selection of the recipients, donor screening, collection and preparation of feces, establishment of a stool bank, frequency and amount of fecal transplant, and efficacy and safety of FMT, as well as other factors in FMT.

### 4.1. The Selection of Recipients

Of all seven studies, the recipients were horses with various gastrointestinal symptoms, including diarrhea, antibiotic-induced intestinal dysbiosis, colitis, and free fecal water. Among them, FMT was totally ineffective in treating horses (n = 10) with free fecal water. In this case, fecal water syndrome did not cause gut microbiota changes in horses; consequently, replacing the intestinal microbiota was unproductive. Indeed, although treating gastrointestinal disorders with FMT seems logical, its effectiveness is dependent on the presence of intestinal dysbiosis. Therefore, it is crucial to choose the right patient for FMT.

### 4.2. The Screening Process for Donors

Currently, the selection process for donors prioritizes safety by avoiding as many risk factors as possible to obtain relatively “healthy” fecal matter. The idea of a “healthy” gut microbiota has yet to be clearly defined in horses, or even in humans. The main goal at present is to enhance the effectiveness of FMT treatment. As shown in Table 1, the number of donor horses ranged from one to three, which may be due to the availability of donor horses rather than frequency or amount of FMT. Of all seven studies, the donor horse(s) went through a screening process. The first step in all the studies was to conduct a physical examination to ensure the donors were clinically healthy, although these examinations varied among the studies (except for two studies done by the same author). The primary purpose was to make certain that the donor had a relatively healthy gut microbiota free from diseases and antimicrobial usage for at least three months. Then, five of seven studies conducted secondary examination to further confirm the safety of the feces, which included testing for coronavirus, *Clostridium difficile* toxins A and B, *Clostridium perfringens* antigens, *Lawsonia intracellularis*, *Neorickettsia risticii* and *Salmonella* sp., and fecal egg count. Although the current donor screening process for equine FMT is considered safe, there might be more efficient and specific methods for donor selection.

### 4.3. Methods for Fecal Collection and Preparation

Collecting feces directly from the rectum is the most popular method (4/7 studies), and produces the freshest and least contaminated fecal samples. However, the biggest drawback is the relatively small amount of feces collected each time. Therefore, this method is more suitable for fecal molecular analysis. Alternatively, using a fecal collector or collecting from the bed immediately is another option; however, it may diminish the viability of the fecal bacteria, especially anaerobic bacteria, if the stool is exposed to the environment for too long. According to a previous study [15], the equine stool can be kept for up to 6 h at room temperature without significant impact on the bacterial composition, but resident microbial population alters after that. Therefore, the storage condition of fecal material is very essential in successful FMT. Human studies have demonstrated that frozen feces can be just as effective as fresh ones [16]. However, Kinoshita et al. reported that frozen stool was ineffective for preventing metronidazole-induced dysbiosis of equine gut microbiota. The authors believe that this result was mainly due to conducting FMT without discontinuing metronidazole administration. Preparing and using frozen feces in horses is worth further exploration as it has significant implications for veterinary practices. Pre-screened frozen feces are more practical in terms of cost and time and allow for greater accessibility in equine clinical practices, which can overcome geographical limitations. It should be noted that appropriate storage conditions are necessary to maintain the viability of the microbial population.

### 4.4. Stool Bank Establishment

Establishing a stool bank for equine fecal samples could serve as an initial step towards implementation of FMT in the future. In addition, horses are highly admired for their athletic abilities such as jumping and running, unlike other animals. Recent evidence suggests that the gut microbiota plays a crucial role in human performance [17,18]. Studies have revealed that a higher abundance of lactic acid-utilizing bacteria in the gut is associated with improved sport performance. Although there is a dearth of research on this topic, an in vitro study identified the presence of lactate-utilizing bacteria in the equine gut microbiota community [19]. While gut microbiota is not a predictor of performance in endurance races in horses [20], it is plausible that lactic-acid utilizing bacteria in the intestinal tract can enhance their athletic abilities. Thus, a stool bank using samples from high-performance athlete horses with these bacteria could be utilized as a natural performance booster in sports events.

Obesity is a rising concern among horses as it is associated with metabolic disorders such as insulin imbalances, high lipid levels, and laminitis [21,22,23]. Studies have demonstrated that overweight horses have alterations in their gut microbiota following weight loss, resulting in a significant increase in the alpha-diversity of their fecal microbiota [24]. Considering these findings and the impact of gut microbiomes on fitness, utilizing lean horse feces selected based on Body Condition Score (BCS) as a treatment option for weight loss in overweight horses may be a cost-effective and safe approach.

### 4.5. The Frequency and Amount for Fecal Transplant

The frequencies of FMT were varied among studies, with some using a single FMT, or administration over three or five consecutive days. The volume of stools was also different between studies, from 0.5 kg to 2 kg. These findings suggest that the frequency and amount of FMT are not related to a specific disorder or the weight of the patient, but are instead determined by the veterinary expert.

### 4.6. Efficacy and Safety of FMT

In clinical treatments, FMT horses typically do not receive any pre-treatment (n = 5), except for study purposes (n = 2). However, research on humans and mice has demonstrated that the efficacy of FMT can be enhanced through the use of antibiotics before the procedure [25,26,27]. Commensal bacteria in the gastrointestinal tract can act as a protective barrier, preventing other microbiomes from residing in the gut. Antibiotic treatment prior to FMT aims to disrupt the recipient’s gut microbiota and increase colonization efficacy. However, the use of antibiotics in horses can cause severe consequences such as colitis [28,29,30,31], diarrhea [32,33], colic [34], laminitis [35], etc. Therefore, polyethylene glycol (PEG 4000) could be an alternative choice for eradicating the recipient horse’s gut microbiota as studies on humans and mice have demonstrated the effectiveness of PEG in cleaning the bowel and reducing the microbiome [36,37]. While FMT has been successful in treating horses without pre-treating the recipient’s gut microbiome, exploring the efficacy of PEG in equine FMT could increase effectiveness, reduce the need for repeated treatments, and improve equine welfare. Among the seven studies, excepting one study which did not report the composition of gut microbiota, four studies indicated no significant alterations of gut microbiota after FMT whereas the results of the remaining two studies had obvious changes. Hence, based on the current literature, it is challenging to provide a conclusive answer as to whether the efficacy of FMT could be proved by altered composition of gut microbiota in the recipient. From a practical perspective, the alleviation of clinical symptoms may serve as a more visible means of validating the efficacy of FMT.

### 4.7. Other Factors in FMT

The main purpose of FMT is to restore the disrupted gut microbiota. However, some studies suggest that the efficacy of FMT may also depend on other factors such as fungi and viruses. For example, in humans, a previous study indicates that fungi might have a potential influence on FMT efficacy in recurrent CDI treatment [38]. However, the impact of fungi and viruses on efficiency of FMT treatment is an area requiring further research in veterinary science. In addition, side-products of the gut microbiota, including antimicrobials and secondary bile acid, may play a crucial role in FMT efficacy. The production of antimicrobials, bacteriocin, is directed by the gut microbiota [39]. If the gut microbiota is imbalanced, it can decrease the production of bacteriocins which are responsible for preventing harmful agents from growing and spreading. However, by transplanting microbiota from a healthy donor, bacteriocin production could be restored, leading to the effective elimination of pathogenic and opportunistic microorganisms. Gut microbiota also regulates the production of secondary bile acid [40], which can be altered due to the modified composition of their gut microbiota by FMT. Research has shown that FMT can also restore the Firmicutes phylum and secondary bile acid metabolism in CDI patients, which may prevent the growth and germination of C. difficile spores both in vitro and in vivo [41,42]. In summary, while multiple factors contribute to the effectiveness of FMT in horses, the gut microbiota remains a crucial factor in the treatment’s success.

## 5. Conclusions

Fecal microbiota transplantation (FMT) has emerged as a promising treatment option for various gastrointestinal disorders in horses. This systematic review provides an overview of the current applications of FMT in horses and summarizes some of the key findings from the available literature.

One of the main findings of the review was that FMT appears to be effective in treating certain gastrointestinal disorders in horses, including colitis and diarrhea, with reported success rates ranging from 75% to 91.3%. However, the authors note that the quality of the evidence is generally suboptimal, with small sample sizes and a lack of control groups, which limits the strength of the conclusions that can be drawn. Future studies should aim to address these limitations by using larger sample sizes, more rigorous study designs, and standardized protocols for administering FMT.

Another important consideration in the use of FMT is donor selection. Most studies used healthy horses as donors, but there is limited evidence on the optimal criteria for selecting appropriate donors. Some potential factors to consider may include the diversity and stability of their microbiota. Future research should aim to establish clear guidelines for donor selection to ensure the safety and efficacy of FMT in horses.

The safety of FMT in horses is another area of concern. While most studies reported no adverse effects associated with FMT, the long-term safety of the procedure is not yet clear. Additionally, the potential risks of FMT, such as the transmission of infectious diseases and the possibility of introducing harmful or unknown microorganisms into the recipient’s gut, highlight the need for more research on safety and risk management strategies.

Finally, while most studies focused on the use of FMT for treating gastrointestinal disorders, there is limited evidence on the potential applications of FMT for other conditions in horses, such as obesity and metabolic disorders. Future research should explore the potential benefits of FMT in these areas and investigate the underlying mechanisms of action.

In conclusion, the systematic review provides a comprehensive overview of the current applications of FMT in horses and highlights the need for more research in this area. While FMT appears to be a promising treatment option for certain gastrointestinal disorders in horses, more rigorous studies are needed to establish the optimal donor selection, dosing, and administration protocols, as well as the long-term safety and efficacy of FMT in horses. Additionally, future research should explore the potential applications of FMT for other conditions in horses and investigate the underlying mechanisms of action.

## Figures and Tables

**Figure 1 vetsci-10-00290-f001:**
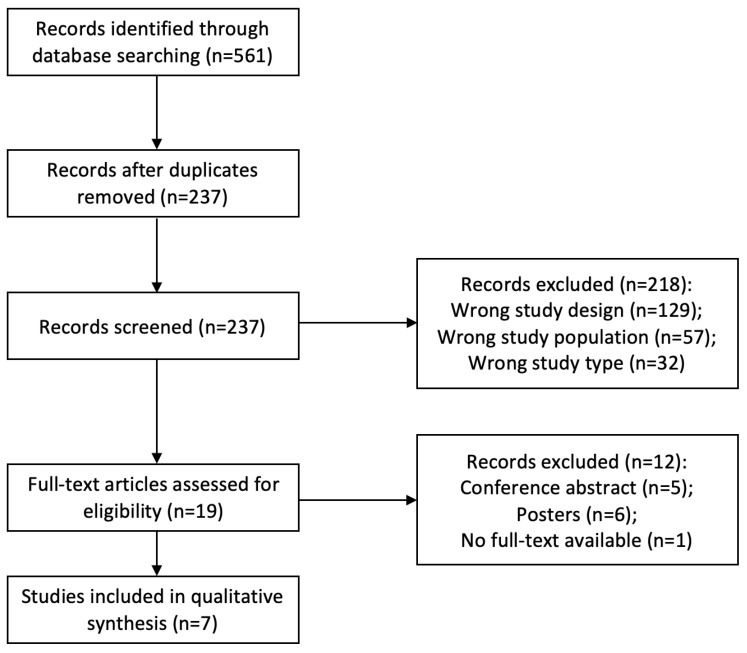
PRISMA flowchart of included studies.

**Table 1 vetsci-10-00290-t001:** Characteristics of the included studies.

Author	Year	Country	Sample Size	Patient Characteristics	FMT Frequency	Fecal Preparation Methods
Dias et al. [9]	2018	Brazil	4	Colitis	Once	1.5–2 kg of fresh stool mixed with 40 g of sodium bicarbonate in 4 L of water
McKinne et al. [10]	2020	USA	5	Diarrhea	3 consecutive days	2.5 pounds of fresh manure mixed with 4 L of lukewarm water
Di Pietro et al. [11]	2021	Canada	9	Antibiotic-induced intestinal dysbiosis	3 consecutive days	1, 1 kg of fresh stool mixed with 2 L of water and centrifuged 2, 1.6 kg of fresh stool mixed with 3.2 L of water
McKinne et al. [8]	2021	USA	22	Diarrhea	3 consecutive days	2.5 pounds of fresh manure mixed with 4 L of lukewarm water
Costa et al. [12]	2021	Canada	6	Diarrhea	Once	1.5 kg of fresh stool mixed with 5 L of warm water
Laustsen et al. [13]	2021	The Netherlands	20	Free fecal water	Once	0.5 kg of fresh stool mixed with 5 L of non-sterile warm saline
Kinoshita et al. [14]	2022	Japan	9	Antibiotic-induced intestinal dysbiosis	5 consecutive days	0.5 kg of fresh stool mixed with 1 L of warm water

**Table 2 vetsci-10-00290-t002:** Donor examinations before and after fecal collection.

Reference	No. of Donor	Before Fecal Collection		After Fecal Collection
		Recorded Information	Exclusion Criteria	Included Examinations
[9]	1	Physical examination, history of infectious diseases; history of antimicrobial therapy in recent 6 months; vaccination and deworming	Not specified	Not specified
[10]	3	Complete diet history, medical history, and physical examination; breed, age, body condition score, heart rate, respiratory rate, rectal temperature, attitude, and borborygmi	Any recent gastrointestinal illness (colic, diarrhea), transport, medical treatment, or dietary supplementation with probiotics	Fecal egg count, coronavirus, *Clostridium difficile* toxins A and B, *Clostridium perfringens* antigens, *Lawsonia intracellularis*, *Neorickettsia risticii*, and *Salmonella* sp.
[11]	1	Breed, age, body weight; history of antimicrobials or other medications in the last 3 months	Not specified	*Salmonella enterica*, *Clostridium perfringens*, *Clostridioides difficile*, and parasitic eggs
[8]	3	Complete diet history, medical history, physical examination; breed, age, body condition score, heart rate, respiratory rate, rectal temperature, attitude, and borborygmi	Any recent gastrointestinal illness (colic, diarrhea), transport, medical treatment, or dietary supplementation with probiotics	Coronavirus, *Clostridium difficile* toxins A and B, *Clostridium perfringens* antigens, *Lawsonia intracellularis*, *Neorickettsia risticii*, *Salmonella* sp., and quantitative fecal egg count
[12]	2	Breed, age; history of antimicrobials or other medications in the last 6 months; history of intestinal diseases; history of deworming	Not specified	*Salmonella enterica*, *Clostridium perfringens*, *Clostridioides difficile* by culture, and negative for parasitic eggs
[13]	2	Health status, history of digestive issues; history of medical treatments in the last 12 months; clinical history (>5 years)	Not specified	Not specified
[14]	1	Breed, age, sex; history of antimicrobials in the last 3 months; history of intestinal issues in the last 3 months	Not specified	*Clostridioides difficile*, *Clostridium perfringens*, and *Salmonella* species by culture methods

**Table 3 vetsci-10-00290-t003:** Changes of the equine gut microbiota compositions in the included studies.

Reference	Pre-FMT	Post-FMT
[9]	NA	NA
[10]	The fecal microbiota was significantly more variable in terms of β-diversity	The fecal microbiota had a higher α-diversity than prior to treatment and was phylogenetically more similar to that of the donor
[11]	The fecal microbiota showed greater representation of the genus *Intestinimonas*, unclassified Lactobacillales, *Lactobacillus*, and *Streptococcus*	Simpson’s index was not significantly different comparing patients to each other
[8]	The fecal microbiota showed lower α-diversity and greater beta *β*-diversity	Horses showed a lower mean UniFrac distance
[12]	The Order Lactobacillales and the genera *Lactobacillus*, *Intestinimonas*, and *Streptococcus* were increased in the microbiota of diarrheic horses	No change in the fecal microbiota
[13]	Compared to healthy controls, the fecal microbiota did not show significant differences	No effect on the fecal microbiota in terms of alpha or beta diversity
[14]	NA	Changes in the ratios of bacterial families were similar between the metronidazole-treated group and the simultaneous metronidazole- and FMT-treated group, notably in the *Clostridiaceae*, *Ruminococcaceae*, and *Enterobacteriaceae*. Differences in fecal bacterial compositions were due mainly to metronidazole administration (*p* = 0.0003), but not to FMT

## Data Availability

Not applicable.

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
