# Peer review of "A Systematic Review of Current Applications of Fecal Microbiota Transplantation in Horses"

_vetsci, 2023, doi:10.3390/vetsci10040290_

Round 1

Reviewer 1 Report

The manuscript A systematic review of current applications of fecal microbiota 1 transplantation in horses is a review regarding the transplantation of microbiota in faecal samples. The review is very interesting and very well presented. Some suggestions:

-some sections of the text are formatted different than the rest of the manuscript

- attectio  to the subtitles

- the discussion maybe by divided by section to not be so long

Author Response

Hi reviewer,

Thanks for your time and advice.

- Checked the format again

- Sorry, I'm not sure about your question

- Discussion was divided into seven sections which were highlighted

Thanks

Reviewer 2 Report

    FMT has become an important stools to treat intestinal diseases especially in human beings. This systematic review evaluated the current literature on the use of FMT in horses, including fecal collection, fecal preparation, donor selection and screening and the efficacy and safety.  However, I have the two questions. Firstly, for examlpe, the author presented three fecal preparation methods, which methods did you make a priority selection. Secondly, What are the criteria for evaluating the effectiveness of FMT. When the gut microbiota that from the donors transplant to the recipients, it can certainly change the intestinal community structure within a certain period of time. Whether the recipient's intestinal microbial community return to its pre-transplant state over time. the author can give a presentation in "Discussion" parts.

   In addition, some details issues  are as follows:

1. line 25-26 Although the authors feel previous research are exactly low, why not try to use another expression.

2.  For SCI papers, numbers less than 10 are in English and numbers greater than 10 are list by Arabic.

Author Response

Hi review,

Thank you for your time and comments.

- There are three different methods described in the included literature for fecal collection. However, we did not make a priority selection as the fecal collection method is not related to the efficacy of FMT rather veterinarian`s personal choice.

- FMT certainly can change the microbial composition of the intestine mostly for a short period of time. Therefore, changes of the gut microbiota is not a criteria for FMT efficacy. Instead, we employed the outcome of the issue that is being treated by FMT as the creteria for evaluating the efficacy of the FMT. For example, no more diarrhea is the criteria for FMT efficacy for treating diarrhea.

- "Low" changed to "suboptimal"

- Numbers less than 10 were written in English and highlighted in the article.

Thanks

Reviewer 3 Report

This report addresses the use of fecal microbiota transplation (FMT) in horses. The topic if highly interested in equine medicine because FMT is a common procedure in equine clinics but there are lack of systematic reviews.

The length and structure are correct, and in general terms, it is effortless to understand. The small number of articles and the sample size included, as the authors explain in their limitations, make it not possible to get strong conclussions. Anyway, the article is very interesting for the equine practitioners and encorage to invetigate more about FMT in horses.

Author Response

Hi reviewer,

Thanks for your time and review.